# Two-Phase Flow Simulation of Tunnel and Lee-Wake Erosion of Scour below a Submarine Pipeline

**Antoine Mathieu \*, Julien Chauchat** **, Cyrille Bonamy and Tim Nagel**

Laboratoire des Ecoulements Géophysiques et Industriels (LEGI), Centre National de la Recherche Scientifique (CNRS), University of Grenoble Alpes, Grenoble-INP, F-38000 Grenoble, France
\* Correspondence: antoine.mathieu@univ-grenoble-alpes.fr

**Abstract:** This paper presents a numerical investigation of the scour phenomenon around a submarine pipeline. The numerical simulations are performed using SedFoam, a two-phase flow model for sediment transport implemented in the open source Computational Fluid Dynamics (CFD) toolbox OpenFOAM. The paper focuses on the sensitivity of the granular stress model and the turbulence model with respect to the predictive capability of the two-phase flow model. The quality of the simulation results is estimated using a statistical estimator: the Brier Skill Score. The numerical results show no sensitivity to the granular stress model. However, the results strongly depend on the choice of the turbulence model, especially through the different implementations of the cross-diffusion term in the dissipation equation between the $k - \varepsilon$ and the $k - \omega 2006$ models. The influence of the cross-diffusion term tends to indicate that the sediment transport layer behaves more as a shear layer than as a boundary layer, for which the $k - \varepsilon$ model is more suitable.

**Keywords:** two-phase flow; scour; pipeline; numerical modeling; turbulence modeling

## 1. Introduction

The scour phenomenon is a major cause of the rupture or self-burying of submarine pipelines. This phenomenon has to be accurately predicted during the design process of these submarine structures. An important amount of experimental and numerical studies has been published in the literature to characterize both the shape and the time development of the scour hole. The erosion under a submarine pipeline can be decomposed into three steps: (1) the onset; when the current around the cylinder is strong enough, it generates a pressure drop, which liquefies the sediments underneath the cylinder; (2) the tunneling stage; when a breach is formed between the cylinder and the sediment bed, it expands due to the strong current in the breach; (3) the lee-wake erosion stage; when the gap is large enough, vortices are shed in the wake of the cylinder, leading to erosion downstream of the scour hole [1]. Mao (1986) [2] performed extensive experiments for which the shape of the sediment bed around the cylinder was recorded for various flow conditions. The data collected by Mao (1986) [2] have been widely used as a benchmark for sediment transport models applied to the scour phenomenon.

The numerical simulation of scour under a pipeline has been extensively investigated during the last few decades. The first numerical simulations were based on single-phase models and differed mainly by the way turbulence was modeled. Models based on potential flow theory like Chiew (1991) [3] or Li and Cheng (1999) [4] tend to predict correctly the final shape of the upstream part of the scour hole and its maximum depth. However, they fail to reproduce the downstream part because, according to Sumer (2002) [1], the erosion in this region is dominated by the wake of the cylinder and vortex shedding in the sediment bed. Other single-phase flow models using $k - \varepsilon$ turbulence models like Leeuwenstein and Wind (1984) [5] were used to determine the shape of the scour hole, but also failed

to reproduce the equilibrium stage. Indeed, in early numerical models, the role played by the suspended sediment was underestimated, and only the bed load transport component of sediment transport was taken into account. Clear water cases were better predicted than live-bed cases because suspension plays a more important role in live-bed conditions. Liang et al. (2004) [6] presented a numerical model able to simulate the time development of the scour hole in live-bed and clear-water conditions. The authors' model took into account bed load and suspended load transport, and a $k - \varepsilon$ turbulence model was used. Furthermore, the influence of the local bed slope on the critical shear stress was incorporated. This means that the threshold Shields parameter $\theta_c$ was adjusted to a higher value for sediments moving up slope and to lower value for sediments moving down slope. The mathematical model presented by Liang et al. (2004) [6] showed fairly good agreement with the experimental data from Mao (1986) [2], but the accumulation of sediment behind the pipeline was over-predicted for both live-bed and clear-water scour cases. Liang et al. (2004) [6] argued that this over-prediction was caused by the choice of the $k - \varepsilon$ turbulence model. Indeed, the $k - \varepsilon$ model tends to smooth out the fluctuations in the wake of the pipeline, and therefore, the interaction with the sediment bed can be altered.

The mutual interactions between the fluid and the sediments are more complex than a simple local shear stress relation, as is classically assumed in single-phase flow models. During the past two decades, a new modeling approach has emerged, two-phase flow models [7–9]. One of the main advantages over classical models is that the two-phase flow approach does not require the use of the empirical sediment transport rate and erosion-deposition formulas. The physical grounds on which this new generation of sediment transport models is based should allow improving scour modeling. The two-phase flow approach has already been applied to the scour below a submarine pipeline configuration. Zhao and Fernando (2007) [10] used a two-phase model implemented in the CFD software FLUENT with a $k - \varepsilon$ turbulence model. The temporal evolution of the maximum scour hole was captured in clear-water conditions, but they found that sediments were still in motion in the "fixed" sediment bed layer. It was the first time that a two-phase flow model was applied to the case of scour under a pipeline. At the time, their results were encouraging given the complexity of the phenomenon.

Bakhtiary et al. (2011) [11] also simulated scour under a pipeline in clear-water conditions with a $k - \varepsilon$ turbulence model. The tunneling stage of scour was correctly reproduced, and the shape of the sediment bed corresponded to the experimental data from Mao (1986) [2]. Nevertheless, the upstream part of the scour hole seemed to be better predicted than the downstream part.

More recently, Lee et al. (2016) [12] used a two-phase flow model with a $k - \varepsilon$ turbulence model and the $\mu(I)$ rheology for the granular phase [13]. The authors performed simulations of scour under a pipeline in live-bed conditions. The undisturbed Shields number was higher than in the previous simulations, and no comparison can be made between this model and the other models cited previously. Nevertheless, the time evolution of the scour hole corresponded to the experimental data. They found a strong influence of the turbulence model parameters on the final morphology. Two-phase flow models reached the level of performance of the classical models. Lee et al. (2016) [12] also pointed out the limitation of the $k - \varepsilon$ model and the necessity to use a $k - \omega$-type turbulence model for prediction of the final shape of the scour hole downstream of the pipeline.

The revisited $k - \omega$ turbulence model from Wilcox (2006) [14] adapted for two-phase flow by Nagel (2019) [15] (referred to as $k - \omega 2006$ in this paper) can reproduce vortex-shedding phenomenon (see Appendix A). It should therefore be able to simulate the lee-wake erosion stage.

In Section 2, the two-phase flow model is presented. The numerical setup is detailed in Section 2.4, and the results are presented and discussed in Section 3.

## 2. Materials and Methods

### 2.1. Governing Equations

The two-phase flow model for sediment transport, SedFoam, developed by Cheng et al. (2017) [16] and Chauchat et al. (2017) [17], was used as a starting point. SedFoam is an open-source and

multi-version solver implemented in the open-source CFD toolbox OpenFOAM and proposes several granular stress and turbulence models (https://github.com/SedFoam/sedfoam). SedFoam is designed to study three-dimensional sediment transport.

In the two-phase flow formulation, both solid and fluid phases are described by Eulerian equations. The fluid and solid mass conservation equations are written as:

$$\frac{\partial \phi}{\partial t} + \frac{\partial \phi u_i^s}{\partial x_i} = 0 \tag{1}$$

$$\frac{\partial (1-\phi)}{\partial t} + \frac{\partial (1-\phi)u_i^f}{\partial x_i} = 0 \tag{2}$$

where $\phi$ is the sediment phase concentration, $u_i^s$ and $u_i^f$ are the sediment and fluid phase velocities, respectively, and $i = 1, 2$ are the stream-wise and vertical components.

The momentum equations for the solid and fluid phases are given by:

$$\frac{\partial \rho^s \phi u_i^s}{\partial t} + \frac{\partial \rho^s \phi u_i^s u_j^s}{\partial x_j} = -\phi \frac{\partial p}{\partial x_i} - \frac{\partial p^s}{\partial x_i} + \frac{\partial \tau_{ij}^s}{\partial x_j} + \phi \rho^s g_i + \phi(1-\phi)K(u_i^f - u_i^s)$$
$$-(1-\phi)\frac{1}{\sigma_c}K\nu_t^f \frac{\partial \phi}{\partial x_i} \tag{3}$$

$$\frac{\partial \rho^f (1-\phi)u_i^f}{\partial t} + \frac{\partial \rho^f (1-\phi)u_i^f u_j^f}{\partial x_j} = -(1-\phi)\frac{\partial p}{\partial x_i} + \frac{\partial \tau_{ij}^f}{\partial x_j} + (1-\phi)\rho^f g_i$$
$$-\phi(1-\phi)K(u_i^f - u_i^s) + (1-\phi)\frac{1}{\sigma_c}K\nu_t^f \frac{\partial \phi}{\partial x_i} \tag{4}$$

with $\rho^s$ and $\rho^f$ the solid and fluid density, $g_i$ the acceleration of gravity, $p$ the fluid pressure, $p^s$ the solid phase normal stress, and $\tau_{ij}^s$ and $\tau_{ij}^f$ the solid and fluid phase shear stresses. The solid phase shear stress closure model is detailed in Section 2.2, and the fluid phase shear stress is expressed as:

$$\tau_{ij}^f = \rho^f (1-\phi)\left[2\nu_{Eff}S_{ij}^f - \frac{2}{3}k\delta_{ij}\right]. \tag{5}$$

$S_{ij}^f = 1/2\left(\partial u_i^f/\partial x_j + \partial u_j^f/\partial x_i\right) - 1/3\left(\partial u_k^f/\partial u_k^f\right)$ is the deviatoric part of the fluid strain rate tensor; $k$ is the turbulent kinetic energy (TKE); and $\nu_{Eff}$ is the effective velocity defined by $\nu_{Eff} = \nu_t^f + \nu_{mix}$ with $\nu_t^f$ the eddy viscosity calculated by a turbulence closure model (see Section 2.3) and $\nu_{mix}$ the mixture viscosity following the model proposed by Boyer et al. (2011) [18]:

$$\frac{\nu_{mix}}{\nu^f} = 1 + 2.5\phi \left(1 - \frac{\phi}{\phi_{max}}\right)^{-1}, \tag{6}$$

where $\phi_{max} = 0.635$ is the maximum value for the solid phase concentration and $\nu^f$ is the fluid kinematic viscosity.

The last two terms of the right-hand side of both momentum equations represent the drag force coupling the two phases. $\sigma_c$ is the Schmidt number, and $K$ is the drag parameter modeled according to Richardson and Zaki (1954) [19]:

$$K = 0.75C_d \frac{\rho^s}{d} \parallel u^f - u^s \parallel (1-\phi)^{-h_{Exp}}. \tag{7}$$

$d$ is the particles' diameter; $h_{Exp}$ is the hindrance exponent controlling the drag increase with increasing solid concentration; and $C_d$ is the drag coefficient calculated by the empirical formula given by Schiller and Naumann (1933) [20]:

$$
C_d = \begin{cases} \dfrac{24}{Re_p}(1 + 0.15 Re_p^{0.687}), & Re_p \leq 1000 \\ 0.44, & Re_p > 1000 \end{cases},
$$ (8)

where $Re_p$ is the particulate Reynolds number defined by: $Re_p = (1 - \phi) \parallel u^f - u^s \parallel d / \nu^f$.

### 2.2. Granular Stress Models

The particle pressure is the sum of the the pressure induced by collision $p^s$, calculated differently depending on the solid phase stress closure model and the pressure induced by the permanent contact between the particles $p^{ff}$ defined as:

$$
p^{ff} = \begin{cases} 0, \phi < \phi_{min}^{Fric} \\ Fr \dfrac{(\phi - \phi_{min}^{Fric})^{\eta_0}}{(\phi_{max} - \phi)^{\eta_1}}, \phi \geq \phi_{min}^{Fric}, \end{cases}
$$ (9)

where $\phi_{min}^{Fric} = 0.57$, $Fr = 0.05$, $\eta_0 = 3$, and $\eta_1 = 5$ are empirical coefficients. In the present work, numerical simulations are conducted using two granular stress models: the dense granular flow rheology ($\mu(I)$ rheology) and the kinetic theory for granular flows (KT).

#### 2.2.1. $\mu(I)$ Rheology

The pressure induced by collisions and frictional interactions is modeled following Chauchat et al. (2017) [21]:

$$
p^s = \left( \frac{B_\phi \, \phi}{\phi_{max} - \phi} \right)^2 \rho^s d^2 \parallel S^s \parallel^2
$$ (10)

where $B_\phi = 1/3$ is a parameter of the dilatancy law [22] and $\parallel S^s \parallel$ is the norm of the deviatoric part of the solid phase strain rate tensor $S_{ij}^s$ defined as $\parallel S^s \parallel = \sqrt{2 S_{ij}^s S_{ij}^s}$.

Following Jop et al. (2006) [23], the particle shear stress is proportional to the particle pressure following a frictional law:

$$
\tau_{ij}^s = \mu(I)(p^s + p^{ff}) \frac{S_{ij}^s}{\parallel S^s \parallel}.
$$ (11)

According to GDRmidi(2004) [13], the friction coefficient $\mu(I)$ is given by:

$$
\mu(I) = \mu_s + \frac{\mu_2 - \mu_s}{I_0 / I + 1},
$$ (12)

where $I = \parallel S^s \parallel d \sqrt{\rho^s / \tilde{p}^s}$ is the inertial number, $\mu_s = 0.63$ is the static friction coefficient for sand, and $\mu_2 = 1.13$ and $I_0 = 0.6$ are empirical coefficients.

A frictional shear viscosity is introduced to be consistent with the fluid phase momentum equation, and the particle shear stress is written as $\tau_{ij}^s = \nu_{Fr}^s S_{ij}^s$, with the frictional shear viscosity $\nu_{Fr}^s$ written as:

$$
\nu_{Fr}^s = min \left( \frac{\mu(I)(p^s + p^{ff})}{\rho^s \left( \parallel S^s \parallel^2 + D_{small}^2 \right)^{1/2}}, \nu_{max} \right).
$$ (13)

$D_{small}$ is regularization parameter from Chauchat and Médale (2014) [24], and $\nu_{max}$ is a viscosity limiter set to $\nu_{max} = 10$

### 2.2.2. Kinetic Theory for Granular Flows

The model adopted was suggested by Ding and Gidaspow (1990) [25]. The particulate pressure is a function of the particle velocity fluctuations represented by the granular temperature $\Theta$ following:

$$p^s = \rho^s \phi [1 + 2(1+e)\phi g_{s0}]\Theta, \tag{14}$$

with $e$ the coefficient of restitution during the collision and $g_{s0} = (2-\phi)/2(1-\phi)^3$ a radial distribution function from Carnahan and Starling (1969) [26] introduced to describe the crowdedness of particles.

The particle shear stress $\tau_{ij}^s$ is decomposed into the sum of a frictional and a collisional stress component:

$$\tau_{ij}^s = \tau_{ij}^{ff} + \tilde{\tau}_{ij}^s. \tag{15}$$

The frictional components allow reproducing the immobile sediment bed behavior and are defined as $\tau_{ij}^{ff} = 2\rho^s \nu_{Fr}^s S_{ij}^s$, with $\nu_{Fr}^s$ calculated as:

$$\nu_{Fr}^s = \frac{p^{ff} sin(\theta_f)}{\rho^s \left(|| S^s ||^2 + D_{small}^2\right)^{1/2}}, \tag{16}$$

using a constant friction angle $\theta_f = 32°$.

The particle collisional stress is calculated as:

$$\tilde{\tau}_{ij}^s = 2\mu^s S_{ij}^s + \lambda \frac{\partial u_k^s}{\partial x_k}\delta_{ij}. \tag{17}$$

The particle shear viscosity $\mu^s$ and bulk viscosity $\lambda$ are functions of the granular temperature and the radial distribution function following:

$$\mu^s = \rho^s d \sqrt{\Theta} \left[ \frac{4\phi^2 g_{s0}(1+e)}{5\sqrt{\pi}} + \frac{\sqrt{\pi} g_{s0}(1+e)(3e-1)\phi^2}{15(3-e)} + \frac{\sqrt{\pi}\phi}{6(3-e)} \right] \tag{18}$$

and:

$$\lambda = \frac{4}{3}\phi^2 \rho^s d g_{s0}(1+e)\sqrt{\frac{\Theta}{\pi}}. \tag{19}$$

The balance equation for the granular temperature is written as:

$$\frac{3}{2}\left[ \frac{\partial \phi \rho^s \Theta}{\partial t} + \frac{\partial \phi \rho^s u_j^s \Theta}{\partial x_j} \right] = (-p^s \delta_{ij} + \tilde{\tau}_{ij}^s)\frac{\partial u_i^s}{\partial x_j} - \frac{\partial q_j}{\partial x_j} - \gamma + J_{int}, \tag{20}$$

where the granular temperature flux $q_j$ is modeled following Fourier's law of conduction:

$$q_j = -D_\Theta \frac{\partial \Theta}{\partial x_j} \tag{21}$$

with $D_\Theta$ the conductivity calculated as:

$$D_\Theta = \rho^s d \sqrt{\Theta} \left[ \frac{2\phi^2 g_{s0}(1+e)}{\sqrt{\pi}} + \frac{9\sqrt{\pi} g_{s0}(1+e)^2(2e-1)\phi^2}{2(49-33e)} + \frac{5\sqrt{\pi}\phi}{2(49-33e)} \right]. \tag{22}$$

The expression of the dissipation rate of granular temperature $\gamma$ is modeled following Ding and Gidaspow (1990) [25]:

$$\gamma = 3(1 - e^2)\phi^2 \rho^s g_{s0}\Theta \left[ \frac{4}{d}\sqrt{\frac{\Theta}{\pi}} - \frac{\partial u_j^s}{\partial x_j} \right]. \tag{23}$$

Finally, the fluid particle interaction term $J_{int}$ is expressed as:

$$J_{int} = \phi K(2\alpha k - 3\Theta), \tag{24}$$

where $\alpha$ characterizes the degree of correlation between particles and fluid velocity fluctuations following the expression $\alpha = e^{-BSt}$, where $B$ is an empirical coefficient and $St$ is the Stokes number defined as the ratio between the particle response time $t_p = \rho^s/(1 - \phi)K$ and the characteristic time scale of the most energetic eddies $t_l = k/(6\varepsilon)$, with $\varepsilon$ the dissipation rate of TKE.

## 2.3. Turbulence Models

Three two-phase flow versions of Reynolds-averaged Navier–Stokes (RANS) turbulence models were used in the simulations: a $k - \varepsilon$ model [9,16], a $k - \omega 2006$ model [15], and a $k - \varepsilon$ turbulence model written in terms of the specific dissipation rate of TKE $\omega$, denoted hereafter as the modified $k - \varepsilon$ model. The last model was only used to compare the $k - \varepsilon$ and $k - \omega 2006$ behaviors.

In the framework of two-equation RANS turbulence models, transport equations for the dissipation rate and for the TKE need to be solved to compute the turbulent viscosity $v_t^f$. The general expression for the TKE transport equation is given by:

$$\frac{\partial k}{\partial t} + u_j^f \frac{\partial k}{\partial x_j} = \frac{R_{ij}}{\rho^f}\frac{\partial u_i^f}{\partial x_j} + \frac{\partial}{\partial x_j}\left[ \left(v^f + \sigma_k v_t^f\right)\frac{\partial k}{\partial x_j} \right] - \varepsilon$$
$$- \frac{2K(1 - \alpha)\phi k}{\rho^f} - \frac{v_t^f}{\sigma_c(1 - \phi)}\frac{\partial \phi}{\partial x_j}(s - 1)g_i \tag{25}$$

with $R_{ij}^f$ the Reynolds stress tensor and $\sigma_k$ an empirical coefficient.

The transport equation for the dissipation rate and the expression of the turbulent viscosity differ for the different turbulence models.

### 2.3.1. $k - \varepsilon$ Model

For the $k - \varepsilon$ model, the turbulent viscosity $v_t^f$ is calculated as:

$$v_t^f = C_\mu \frac{k^2}{\varepsilon}, \tag{26}$$

and the following transport equation for the dissipation rate $\varepsilon$ is solved:

$$\frac{\partial \varepsilon}{\partial t} + u_j^f \frac{\partial \varepsilon}{\partial x_j} = C_{1\varepsilon}\frac{\varepsilon}{k}\frac{R_{ij}}{\rho^f}\frac{\partial u_i^f}{\partial x_j} + \frac{\partial}{\partial x_j}\left[ \left(v^f + \sigma_\varepsilon v_t^f\right)\frac{\partial \varepsilon}{\partial x_j} \right] - C_{2\varepsilon}\frac{\varepsilon^2}{k}$$
$$- C_{3\varepsilon}\frac{\varepsilon}{k}\frac{2K(1 - \alpha)\phi k}{\rho^f} - C_{4\varepsilon}\frac{\varepsilon}{k}\frac{v_t^f}{\sigma_c(1 - \phi)}\frac{\partial \phi}{\partial x_j}(s - 1)g_i. \tag{27}$$

The values of the empirical coefficients $\sigma_k$, $\sigma_\varepsilon$, $C_{1\varepsilon}$, $C_{2\varepsilon}$, $C_{3\varepsilon}$, $C_{4\varepsilon}$, and $C_\mu$ are listed in Table 1.

**Table 1.** Empirical coefficients for the $k - \varepsilon$ turbulence model from Chauchat et al. (2017) [17].

| $\sigma_k$ | $\sigma_\varepsilon$ | $C_{1\varepsilon}$ | $C_{2\varepsilon}$ | $C_{3\varepsilon}$ | $C_{4\varepsilon}$ | $C_\mu$ |
|---|---|---|---|---|---|---|
| 1.0 | 0.77 | 1.44 | 1.92 | 1.2 | 1.0 | 0.09 |

### 2.3.2. $k - \omega 2006$ Model

The dissipation rate $\varepsilon$ can be expressed in term of specific dissipation rate $\omega$ following the expression $\varepsilon = C_\mu k \omega$. The $k - \omega 2006$ turbulence model uses $\omega$ and the norm of the deviatoric part of the strain rate tensor $|| S^f ||$ to compute the eddy viscosity:

$$v_t^f = \frac{k}{max \left[ \omega, \, C_{lim} \dfrac{|| S^f ||}{\sqrt{C_\mu}} \right]} \tag{28}$$

Compared with the $k - \varepsilon$ or the standard $k - \omega$ model, a stress limiter is incorporated and adjusted by the coefficient $C_{lim}$ [14].

The transport equation for the specific dissipation rate $\omega$ reads:

$$\frac{\partial \omega}{\partial t} + u_j^f \frac{\partial \omega}{\partial x_j} = C_{1\omega} \frac{\omega}{k} \frac{R_{ij}}{\rho^f} \frac{\partial u_i^f}{\partial x_j} + \frac{\partial}{\partial x_j} \left[ \left( v^f + \sigma_\omega v_t^f \right) \frac{\partial \omega}{\partial x_j} \right] - C_{2\omega} \omega^2 - C_{3\omega} \omega \frac{2K(1 - \alpha)\phi}{\rho^f}$$
$$- C_{4\omega} \frac{\omega}{k} \frac{v_t^f}{\sigma_c(1 - \phi)} \frac{\partial \phi}{\partial x_j} (s - 1) g_i + \sigma_d \frac{1}{\omega} \frac{\partial k}{\partial x_j} \frac{\partial \omega}{\partial x_j}. \tag{29}$$

The empirical coefficients for this turbulence model are presented in Table 2, and the coefficient before the cross-diffusion term $\sigma_d$ is given by:

$$\sigma_d = \begin{cases} 0 & \text{for} \quad \dfrac{\partial k}{\partial x_j} \dfrac{\partial \omega}{\partial x_j} < 0 \\[3mm] \dfrac{1}{8} & \text{for} \quad \dfrac{\partial k}{\partial x_j} \dfrac{\partial \omega}{\partial x_j} \geq 0. \end{cases} \tag{30}$$

**Table 2.** Empirical coefficients for the $k - \omega 2006$ turbulence model.

| $\sigma_k$ | $\sigma_\omega$ | $C_{1\omega}$ | $C_{2\omega}$ | $C_{3\omega}$ | $C_{4\varepsilon}$ | $C_\mu$ | $C_{lim}$ |
|------|------|------|--------|------|------|------|-------|
| 0.6 | 0.5 | 0.52 | 0.0708 | 0.35 | 1.0 | 0.09 | 0.875 |

### 2.3.3. Modified $k - \varepsilon$ Model

The modified $k - \varepsilon$ model is obtained by substituting the dissipation rate $\varepsilon$ by $C_\mu k \omega$ in Equations (26) and (27). The new expression for the eddy viscosity is given by:

$$v_t^f = \frac{k}{\omega} \tag{31}$$

and the new dissipation rate transport equation is the same as Equation (29) from the $k - \omega 2006$ model with different coefficients (see the coefficients in Table 3). The coefficient $\sigma_d$ in front of the cross-diffusion term allows switching from the $k - \varepsilon$ and the $k - \omega 2006$ models and investigating the sensitivity of the model to this cross-diffusion term.

$$\sigma_d = \begin{cases} 0 & \text{for} \quad \dfrac{\partial k}{\partial x_j} \dfrac{\partial \omega}{\partial x_j} < 0 \\[3mm] 1.712 & \text{for} \quad \dfrac{\partial k}{\partial x_j} \dfrac{\partial \omega}{\partial x_j} \geq 0. \end{cases} \tag{32}$$

**Table 3.** Empirical coefficients for the modified $k - \varepsilon$ turbulence model.

| $\sigma_k$ | $\sigma_\omega$ | $C_{1\omega}$ | $C_{2\omega}$ | $C_{3\omega}$ | $C_{4\varepsilon}$ | $\sigma_d$ |
|------|-------|------|--------|------|------|----------------------|
| 1.0 | 0.856 | 0.44 | 0.0828 | 0.35 | 1.0 | 1.712 or Equation (32) |

*2.4. Numerical Setup*

Following Lee et al. (2016) [12], the configuration of Mao (1986) [2] was used as a benchmark to study the time evolution of the bed morphology. A pipeline having a diameter $D = 5$ cm was placed just above a sediment bed made of medium sand (median diameter $d_{50} = 360$ μm and density $\rho^s = 2600$ kg·m$^{-3}$). The incoming current had a mean velocity $\overline{U} = 0.87$ m·s$^{-1}$ corresponding to a Shields number $\theta_\infty = 0.33$. Initially, the pipeline was just laid on the sediment bed with no embedment. Mao (1986) [2] measured the sediment bed profile at different times until scour equilibrium was reached. For the simulations, the dynamics was resolved up to 30 s to ensure that the small perturbations of the sediment bed coming from the inlet did not reach the scour hole.

2.4.1. General Setup

The numerical domain dimensions presented in Figure 1 are similar to the ones used by Lee et al. (2016) [12]. For both configurations, the cylinder was placed 15 diameters away from the inlet. The overall domain dimensions were 35 diameters long and 6.1 diameters high. The top boundary condition was a symmetry plane. For the reduced pressure, the outlet boundary condition was a homogeneous Dirichlet condition ($p - \rho^f gy = 0$ Pa). For the outlet velocity, a homogeneous Neumann boundary condition was used for outgoing flows, and a homogeneous Dirichlet boundary condition was used for incoming flows. The inlet was decomposed into two parts. From the bottom to $y = 1.5D$, a wall-type boundary condition was applied. From $y = 1.5D$ to the top, a rough wall log law velocity profile was used following the expression:

$$u_1^f(y) = \frac{u_*}{\kappa} ln \left( \frac{30y}{k_s} \right),$$ (33)

where $\kappa = 0.41$ is the von Karman constant, $u_*$ is the friction velocity, and $k_s = 2.5d$ is the Nikuradse roughness length. The different boundary conditions can be found in the test case 2DPipelineScour, publicly available on GitHub (after the paper is accepted). Second order schemes (Gauss linearUpwind) and the default preconditioned biconjugate gradient pressure solver were used for all the simulations presented in this paper.

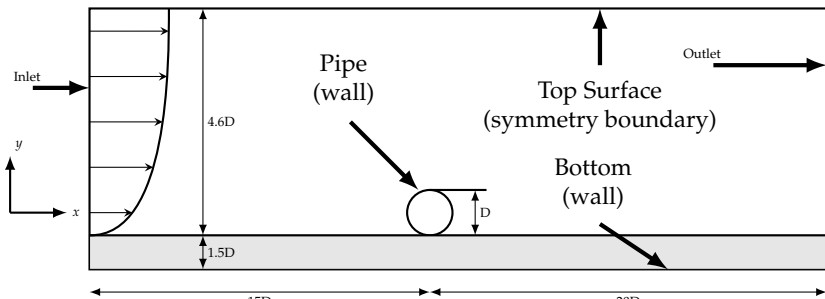

**Figure 1.** Sketch of the geometry and the boundary conditions used for the computational domain.

For the two configurations, following Chauchat et al. (2017) [17], the turbulent parameter $B$ was set to $B = 1$. According to Van Rijn (1984) [27], the value of $\sigma_c$ depends on the suspension number $w_s/u_*$ with $w_s$ the particles' fall velocity:

$$\frac{1}{\sigma_c} = 1 + 2 \left[ \frac{w_s}{u_*} \right]^2, \; 0.1 < \frac{w_s}{u_*} < 1$$ (34)

Therefore, $\sigma_c$ is bounded between 1/3 and one. For the present configuration, the sediment transport was intense; the suspension number was small; and following Lee et al. (2016) [12], $\sigma_c$ was set to one.

The mesh was generated using the OpenFOAM utility snappyHexMesh. Cells were refined in the sediment bed region. Non-refined cells were squares having $3 \times 10^{-3}$m sides, and refined cells were squares having $7.5 \times 10^{-4}$ m sides. The different turbulence models required a specific near-wall resolution. Therefore, cells' refinement close to the cylinder and turbulent boundary conditions depended on the choice of the turbulence model. More details are available in the test case 2DPipelineScour.

### 2.4.2. Simulations with the $k - \varepsilon$ Turbulence Model

The first cells near the cylinder were $6 \times 10^{-4}$ m thick, giving a dimensionless near-wall cell thickness equal to the required $y^+ = 30$ with $y^+$ the dimensionless wall distance defined as $y^+ = u^* y / \nu$. A homogeneous Dirichlet boundary condition of $1 \times 10^{-10}$ m$^2$ s$^{-2}$ was applied on the cylinder surface for $k$, and a homogeneous Neumann boundary condition was applied for the rate of dissipation of the TKE $\varepsilon$. Similar to Lee et al. (2016) [12], inlet values for turbulent quantities were set following Ferziger (2002) [28]: $k = 10^{-4} \bar{U}$ and $\varepsilon = k^{3/2}/0.1$ h, with h the distance from the bed to the top boundary.

### 2.4.3. Simulations with the $k - \omega 2006$ and the Modified $k - \varepsilon$ Turbulence Models

For the simulations using the $k - \omega 2006$ and the modified $k - \varepsilon$ turbulence models, cells near the cylinder were $2 \times 10^{-5}$ m thick, giving a near-wall cell thickness equal to $y^+ = 1$. Wall functions for smooth walls were applied on the cylinder surface for $k$ and $\omega$. Inlet values for $k$ and $\omega$ were calculated similarly to Section 2.3.1 following Ferziger (2002) [28] with $\omega = \varepsilon / C_\mu k$, except that the dissipation was enhanced at the inlet by two orders of magnitude to reduce the incoming TKE, susceptible to damping the vortex-shedding.

### 2.5. Brier Skill Score

In order to estimate the prediction capability of the different combinations of models objectively (granular stress and turbulence models), the Brier Skill Score (*BSS*) was used. It is a statistical approach used to measure the quality of an agreement between simulation results and experimental data. This statistical tool has been extensively used in the coastal engineering community [29–31]. It provides an estimation of a model's performance and is defined following the expression:

$$BSS = 1 - \frac{\sum_i^n |y_i^s - y_i^e|^2}{\sum_i^n |y_i^0 - y_i^e(x)|^2}. \tag{35}$$

The *BSS* compares the sum of the squared difference between the bed elevation $y^s$ from simulations and $y^e$ from experiments at point $i$ (from 0–$n$, the total number of experimental points) with the mean squared difference between the initial bed elevation $y^0$ and $y^e$ also at point $i$. The bed elevation $y^s$ in the simulations is the line of isoconcentration $\phi = 0.5$. A *BSS* equal to one expresses a perfect agreement between simulation and experimental results. The agreement quality decreases with the *BSS*, whereas a negative *BSS* expresses simulation results further away from the experimental results than the initial bed elevation.

## 3. Results

### 3.1. Solid Phase Stress Model Sensitivity

Two simulations were conducted to evaluate the sensitivity of the model results to the choice of the granular stress model: the kinetic theory for granular flows and the $\mu(I)$ rheology. For both

simulations, the $k - \varepsilon$ turbulence model was used. The time evolution of the maximum scour depth and the shape of the sediment bed are compared with the experimental data from Mao (1986) [2] and numerical simulation results from Lee et al. (2016) [12] in Figures 2 and 3, respectively.

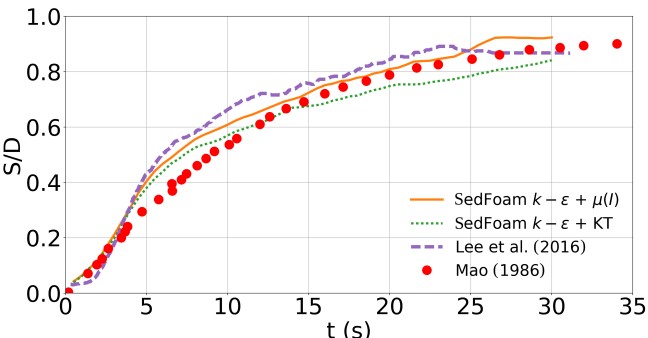

**Figure 2.** Time evolution of the maximum scour depth for simulations with the $k - \varepsilon$ turbulence model using $\mu(I)$ rheology (orange line) and kinetic theory (green dotted line) compared with the experimental data from Mao (1986) [2] (red dots) and numerical data from Lee et al. (2016) [12] (purple dashed line).

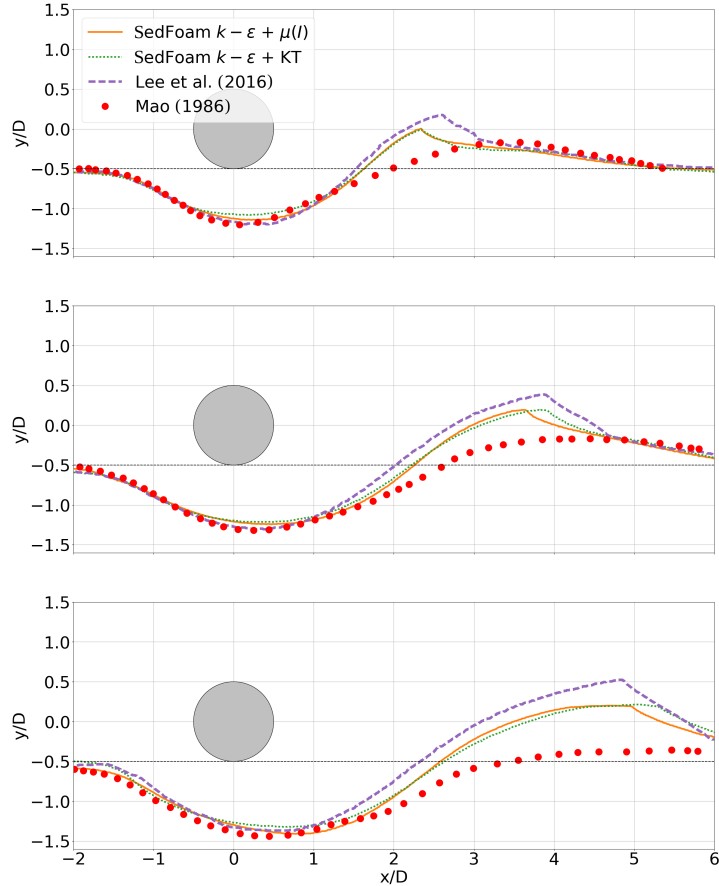

**Figure 3.** Bed profiles from simulations with the $k - \varepsilon$ turbulence model using $\mu(I)$ rheology (orange line) and kinetic theory (green dotted line) at 11 s (top), 18 s (middle), and 25 s (bottom) compared with experimental data from Mao (1986) [2] (red dots) and numerical data from Lee et al. (2016) [12] (purple dashed line).

It clearly appears that the shape of the sediment bed and its time evolution were not very sensitive to the granular stress model. For both models, the temporal evolution of the maximum scour depth (Figure 2) and the upstream part of the sediment bed (Figure 3) agreed reasonably well

with the experimental data from Mao (1986) [2] and the numerical results from Lee et al. (2016) [12]. At 25 s, the *BSS* calculated from the simulations using the $\mu(I)$ rheology ($BSS = 0.731$) and the KT ($BSS = 0.721$) were very close. The quality of the results was therefore equivalent for both granular stress models.

In both simulations, the sediments tended to accumulate at the downstream side of the pipeline, generating a sand dune. According to Lee et al. (2016) [12], this accretion phenomenon can be explained by the inability of the $k - \varepsilon$ model to reproduce the oscillatory wake behind the cylinder (responsible for the lee-wake erosion stage).

The present results demonstrated that the sediment accumulation observed at the lee side of the cylinder was not due to the solid stress model. Since the granular stress models provided similar results, in the following, only the $\mu(I)$ rheology will be used, as it was more computationally efficient.

### 3.2. Turbulence Model Sensitivity

In this subsection, the influence of the turbulence model on the shape of the sediment bed is investigated. The results using the $k - \varepsilon$ and the $k - \omega 2006$ turbulence models are compared in Figures 4 and 5 in term of the time evolution of the maximum scour depth and the shape of the sediment bed.

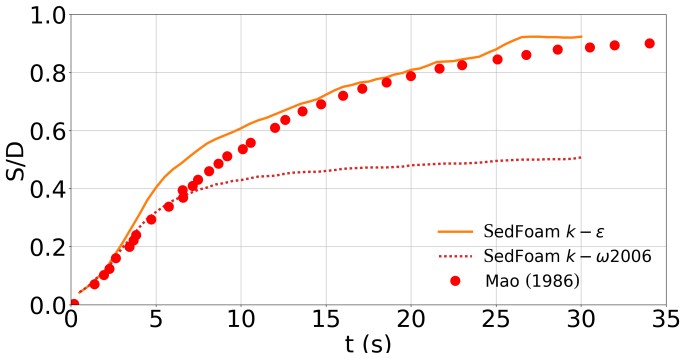

**Figure 4.** Time evolution of the maximum scour depth from simulations with the $\mu(I)$ rheology using the $k - \varepsilon$ (orange line) and $k - \omega 2006$ (red dotted line) turbulence models compared with the experimental data from Mao (1986) [2] (red dots).

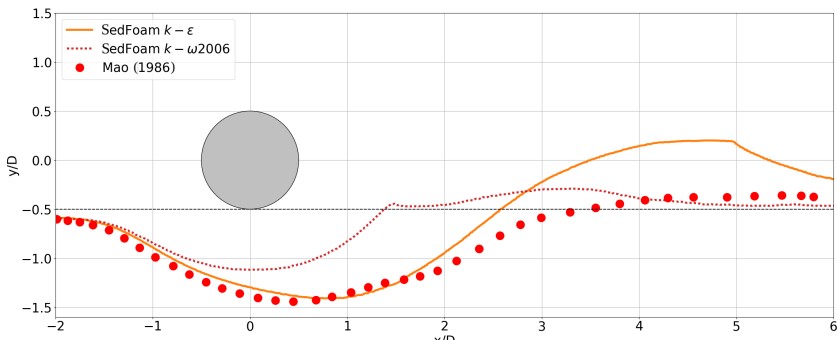

**Figure 5.** Bed profiles at 25 s from simulations with the $\mu(I)$ rheology using the $k - \varepsilon$ (orange line) and $k - \omega 2006$ (red dotted line) turbulence models compared with the experimental data from Mao (1986) [2] (red dots).

The time evolution and the equilibrium depth of the scour hole were significantly underestimated when using the $k - \omega 2006$ turbulence model. This turbulence model did not provide quantitative results in this configuration. However, the vortex shedding phenomenon was predicted, and the sediment bed downstream of the pipeline was eroded (see bed interface between x/D = 4 and x/D = 6).

The snapshot provided in Figure 6 confirms that the erosion was caused by the vortices in the wake of the cylinder. A strong sediment flux was associated with a vortex reaching the sand dune downstream of the pipeline. Therefore, the accretion of sediment visible using the $k - \varepsilon$ model was no longer present when using the $k - \omega 2006$ turbulence model.

The *BSS* at 25 s using the $k - \omega 2006$ was 0.569, which was significantly lower than the one obtained with the $k - \varepsilon$ model. The lee-wake erosion of the sand dune did not compensate the underestimation of the scour depth in the *BSS*.

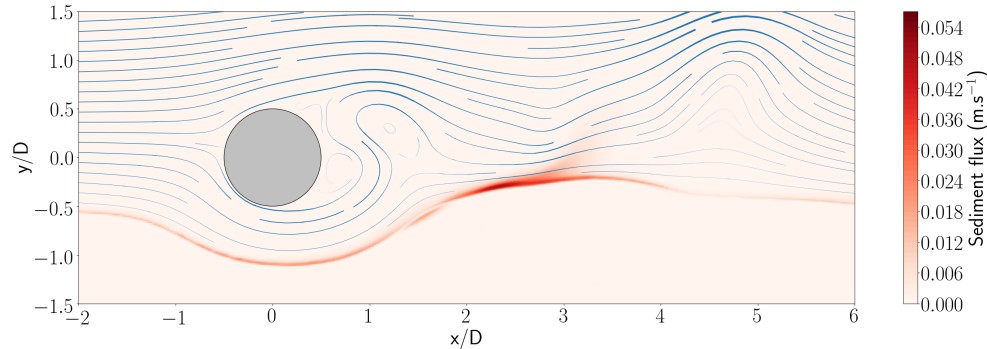

**Figure 6.** Streamlines and sediment volumetric flux at 25 s for the simulation using the $k - \omega/2006$ turbulence model.

A sensitivity analysis on the cross-diffusion term appearing in the dissipation equation through the coefficient $\sigma_d$ was performed to identify the main differences between the two aforementioned models. The time evolution of the maximum scour depths from simulations using the $k - \omega 2006$, the modified $k - \varepsilon$, and the modified $k - \varepsilon$ taking only the positive contribution of the cross-diffusion term is presented in Figure 7. It appears that removing the negative contribution of the cross-diffusion term in the modified $k - \varepsilon$ turbulence model provided results closer to the ones obtained using the $k - \omega 2006$ turbulence model with an equilibrium erosion depth largely underestimated. The definition of the turbulent viscosity in the $k - \omega 2006$ turbulence model mainly affected the dilute regions, but the differences in terms of bed elevation visible in Figures 4 and 7 came from the cross-diffusion term.

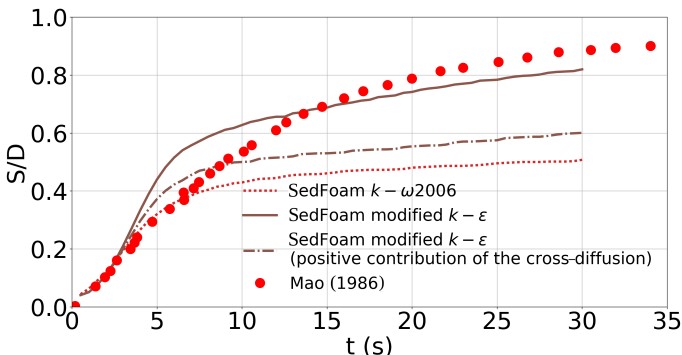

**Figure 7.** Time evolution of the maximum scour depth from simulations with $\mu(I)$ rheology using the $k - \omega 2006$ (red dotted line) and modified $k - \epsilon$ turbulence model with (brown line) and without (brown dashed dotted line) the negative contribution of the cross-diffusion term compared with the experimental data from Mao (1986) [2] (red dots).

The cross-diffusion term was responsible for the behavior of $k - \varepsilon$. It became positive in free shear flows where the $k - \varepsilon$ model is known to provide better predictions and became negative near solid boundaries where the classical $k - \omega$ model provides better predictions. In the $k - \omega 2006$ turbulence model, from Equation (30), only the positive contribution of the cross-diffusion term was incorporated

with a coefficient one order of magnitude smaller than in the $k - \varepsilon$ model. The idea behind the $k - \omega 2006$ model is to incorporate the cross-diffusion term in free shear flows (positive contribution) to have a $k - \varepsilon$ behavior and suppress it near solid boundaries (negative contribution) to have a classical $k - \omega$ behavior.

## 4. Discussion

Close to the sediment bed, the cross-diffusion term became negative. Its contribution was incorporated in the $k - \varepsilon$ model, but not in the $k - \omega 2006$ turbulence model. However, from Figure 7, the negative contribution of the cross-diffusion term played an important role in the time development of the scour hole depth.

The negative contribution of the cross-diffusion term seemed to be necessary to reproduce quantitatively the time development of the scour hole. Typical TKE ($k$) and the specific dissipation rate of the TKE ($\omega$) profiles for free shear, boundary layer, and sediment transport flows are presented in Figure 8. For free shear flows, the peak value of $k$ corresponds to the peak value of $\omega$. The cross-diffusion term was always positive, and the $k - \omega 2006$ turbulence model behaved like a $k - \varepsilon$. For boundary layer flows, the peak value of $\omega$ was located at the wall, whereas the peak value of $k$ was located further away. The gradient of $k$ changed sign toward the boundary, as did the cross-diffusion term, which became negative. The negative cross-diffusion contribution was suppressed in the $k - \omega 2006$ model, having the effect of relaminarizing the flow close to the wall. When sediment transport was involved, the peak values of $k$ and $\omega$ were offset, so that the cross-diffusion term became negative between the two peaks. Using the $k - \omega 2006$ model in this configuration suppressed the influence of the negative contribution of the cross-diffusion term, and the flow was relaminarized close to the sediment bed. This phenomenon was not physical and was responsible for the underestimation of the sediment erosion observed using the $k - \omega 2006$ turbulence model. Finally, our numerical results suggested that sediment transport shares more similarities with a free shear flow than with boundary layer flows. The negative contribution of the cross-diffusion term should therefore be incorporated to behave like a $k - \varepsilon$ model near the sediment bed, while suppressed far from the bed to behave like the $k - \omega 2006$ model and allow vortex-shedding to develop.

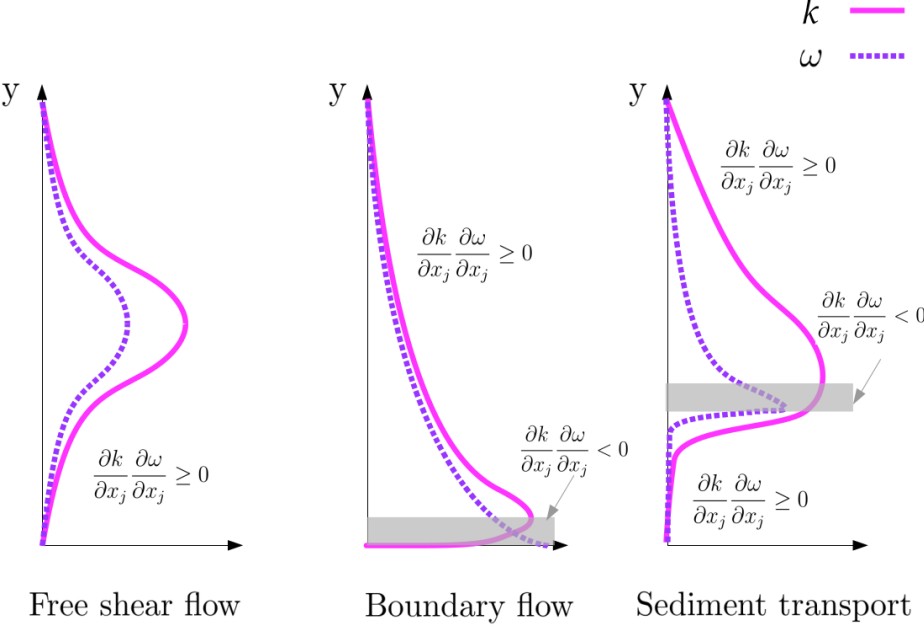

**Figure 8.** Typical turbulent kinetic energy ($k$) and specific dissipation rate ($\omega$) profiles for free shear flows, boundary flows, and sediment transport configurations.

Finally, even though the two-phase flow model relied on a more theoretical background than the classical single-phase flow models, empirical expressions are still needed, especially for the granular stress and turbulence models. However, these models are at a lower level of approximation in the sense that they have been developed and validated on other fluid and granular flow configurations. In this respect, they are more general and better describe the complex physics at work in sediment transport. From Section 3.1, the empiricism in the granular stress model did not seem to be a limitation since the dense granular flow rheology and the kinetic theory of granular flows provided accurate results. However, the available two-phase turbulence models did not fully take into account the complex interactions between the granular phase and the fluid turbulence. For this type of configuration, the coupling between the fluid turbulence and the sediment dynamics was crucial, and Reynolds averaged two-phase flow models showed their limitations.

## 5. Conclusions

This paper presented a numerical investigation of the scour phenomenon below a submarine pipeline. SedFoam, a two-phase flow model for sediment transport applications, was used to study the sensitivity of the scour hole formation and of the bed morphology to the granular stress and the turbulence closure. The quality of the different simulations was measured using the Brier Skill Score. The granular stress model was not sensitive, and similar results were obtained between simulations using $\mu(I)$ rheology and the kinetic theory for granular flows. Both models provided a quantitative time evolution of the erosion depth and of the bed morphology when coupled with the $k - \varepsilon$ turbulence model.

The turbulence model however had a significant influence on the bed morphology. On the one hand, the $k - \varepsilon$ model provided the right equilibrium maximum erosion depth, but overestimated the bed elevation downstream of the pipeline. This accretion phenomenon was explained by the incapacity of the $k - \varepsilon$ model to reproduce the vortex-shedding phenomenon and the lee-wake erosion stage of scour. Therefore, a turbulence model able to reproduce vortex shedding should be used. The $k - \omega 2006$ model, which can reproduce the vortex-shedding, strongly underestimated the erosion depth, but allowed qualitatively reproducing the lee-wake stage of scour.

An in-depth analysis of the $k - \varepsilon$ and the $k - \omega 2006$ revealed the importance of the cross-diffusion term responsible for the behavior of $k - \varepsilon$. The negative and positive contribution of the cross-diffusion term were incorporated in the $k - \varepsilon$ model, whereas only the positive contribution was incorporated in $k - \omega 2006$. The numerical results showed that the negative contribution of the cross-diffusion term was required near the sediment bed to reproduce quantitatively the time development of the scour hole.

An improved URANS two-phase flow turbulence model should have a $k - \omega 2006$ behavior in the outer regions and a $k - \varepsilon$ behavior near the sediment bed. Such a turbulence model would allow providing accurate results in conditions where the interactions between the fluid vortices and the sediment bed are important.

The coupling between the sediments dynamics and the turbulence is a very complex phenomenon, and it should be investigated in detail using large eddy simulations. It would allow better understanding the interactions between the turbulent wake of the cylinder and the sediment bed downstream of the pipeline. This is beyond the scope of the present paper, and it is left for future research.

**Author Contributions:** Funding acquisition, writing—review, project administration and supervision, J.C.; methodology, software, validation, A.M., T.N., C.B. and J.C.; writing—original draft preparation A.M. and J.C.

**Funding:** The work presented in this manuscript was financially supported by the French national research agency ANR projects SegSed (ANR-16-CE01-0005-03) and SheetFlow (ANR-18-CE01-0003) and the European Community's Horizon 2020 Program through the Integrated Infrastructure Initiative HYDRALAB + FREEDATA (654110).

**Acknowledgments:** We would like to thank Cheng-Hsien Lee, Tian-Jian Hsu, and Zhen Cheng for fruitful discussions around multiphase flow modeling of scour. Most of the computations presented in this paper were performed using the GENCI infrastructure under Allocation A0060107567 and the GRICAD infrastructure. We are also grateful to the developers involved in OpenFOAM.

**Conflicts of Interest:** The authors declare no conflict of interest.

## Appendix A. Hydrodynamic Simulations

Two numerical simulations without solid phase ($\phi = 0$) were performed to study the behavior of the turbulence models in the case of a cylinder placed in a steady current. The first simulation used the $k - \omega 2006$ turbulence model presented in Section 2.3.2, and the second used the standard $k - \varepsilon$ turbulence model presented in Section 2.3.1.

*Appendix A.1. Numerical Setup*

The mesh used in both simulations was a 0.41 m (8.2$D$)-wide and 1.2 m (24$D$)-long rectangle with a cylinder of diameter $D = 0.05$ cm placed 0.2 m (4$D$) from the inlet (Figure A1). The top and bottom boundary conditions were symmetry-type boundary conditions. The outlet boundary condition for the reduced pressure was a homogeneous Dirichlet condition ($p - \rho^f gy = 0$ Pa). For the outlet velocity, a homogeneous Neumann boundary condition was used for outgoing flows, and a homogeneous Dirichlet boundary condition was used otherwise. A fixed value of $U = 0.87$ m·s$^{-1}$ was given for the inlet velocity, giving a Reynolds number $Re = UD/\nu = 4.35 \times 10^4$.

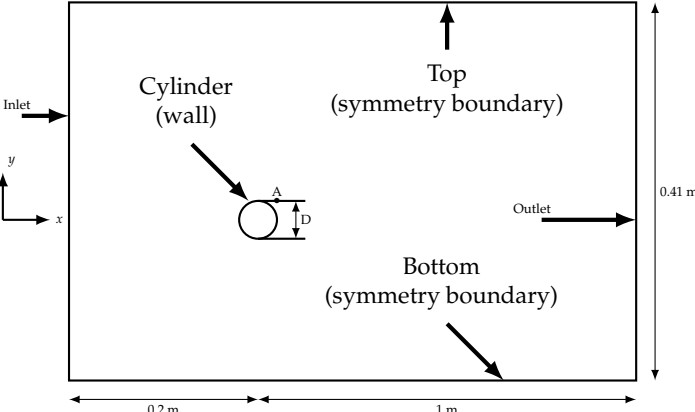

**Figure A1.** Sketch of the geometry and the boundary conditions used for the computational domain.

For the simulation using a $k - \omega 2006$ turbulence model, the cells near the cylinder were $2 \times 10^{-5}$ m thick, giving a near-wall $y^+ \approx 1$. Wall functions for smooth walls were applied on the cylinder surface.

For the simulations using the standard $k - \varepsilon$, the cells near the cylinder were $6 \times 10^{-4}$ m thick, giving a near-wall $y^+ \approx 30$. A homogeneous Dirichlet boundary condition of $1 \times 10^{-10}$ m$^2$·s$^{-2}$ was applied on the cylinder surface for $k$, and a homogeneous Neumann boundary condition was applied for $\varepsilon$.

For both simulations, inlet boundary conditions for turbulent quantities were set following Table A1 with a turbulence intensity $I = 2\%$ and a turbulence length scale $l = 0.07D$.

**Table A1.** Inlet boundary conditions for turbulent quantities.

|  | **Turbulence Kinetic Energy** | **Dissipation** |
|---|---|---|
| $k - \varepsilon$ | $k_{inlet} = \dfrac{3}{2}(UI)^2$ | $\varepsilon_{inlet} = C_\mu \dfrac{k^{1.5}}{l}$ |
| $k - \omega 2006$ | $k_{inlet} = \dfrac{3}{2}(UI)^2$ | $\omega_{inlet} = \dfrac{\sqrt{k}}{l}$ |

Second order differentiation schemes and a time step equal to $2 \times 10^{-4}$ s were used. A probe placed on the top right of the cylinder (Point A in Figure A1) recorded the velocity signal along the simulations.

*Appendix A.2. Results*

The Strouhal number is the dimensionless frequency defined by $St = fD/U$, with $f$ the frequency at which the vortices are shed in the wake of the cylinder. A fast Fourier transform was performed on the velocity signal, and the result is presented in Figure A2. For each simulation, the velocity signal showed one peak value. However, the peak value was not the same for the two turbulence models. Periodic structures were shed in the wake of the cylinder for both simulations, but the frequency differed depending on the turbulence model.

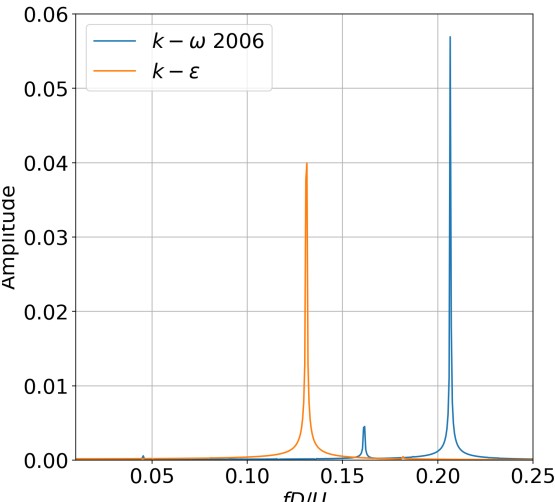

**Figure A2.** Spectrum of the velocity signal.

A typical value of the Strouhal number for cylinder at Reynolds number $Re = 4.35 \times 10^4$ is 0.2 [32]. This value corresponded to the peak value found for the $k - \omega 2006$ turbulence model. The oscillatory behavior of the flow downstream of the pipeline was consistent with the results from the literature. On the contrary, the Strouhal number found with the $k - \varepsilon$ model was largely underestimated. The unsteady turbulent wake behind the cylinder was not properly captured. In conclusion, the $k - \omega 2006$ turbulence model was more suitable to reproduce the lee-wake erosion stage of scour.

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
