# Peer review of "Two-Phase Flow Simulation of Tunnel and Lee-Wake Erosion of Scour below a Submarine Pipeline"

_water, doi:10.3390/w11081727_

Round 1

Reviewer 1 Report

The paper deals with numerical investigation of the scour phenomenon occurring below the submarine pipeline. The authors performed simulations based on different approaches and came up with interesting conclusions. In doing that they also used statistical approach to measure the quality of different simulations. The authors demonstrated that the results are not sensitive to the choice of the granular stress model. On the other hand, it was shown that the turbulence model has a significant influence and the authors have reasonably explained their choice of the turbulence model.

The manuscript is well organized and of interest to the research community. The quality of presentation is adequate. Just a few minor remarks could be considered to improve mainly the style and correct some typos: 

1) Introduction: "... the choice of the turbulence model is very sensitive..." The authors probably want to say that the obtained results are sensitive, i.e. strongly depend on the choice of the turbulence model?

2) Lines 15-21,The sentence beginning with: "The erosion under submarine pipeline can be decomposed into three steps..." is too long. It could be restructured a little bit using "1)", "2)", etc, or ";", or similar. 

3) Line 83: "(the source code will be available via github after the paper is accepted)" This should be reformulated in the final version, should the manuscript get accepted. 

4) Too many "is written", "are written" before equations on page 3. It would be better to vary the style somewhat. 

5) Lines 121 and 126: do not indent the first line after equations, where the quantities used are explained. 

6) Line 214: "m2.s-2" Put the point in the right position (it appears to have a function of a full-stop here)

7) Line 322: Correct "One one hand" to "On one hand".

Author Response

1) Introduction: "... the choice of the turbulence model is very sensitive..." The authors probably want to say that the obtained results are sensitive, i.e. strongly depend on the choice of the turbulence model?

Response: sentence modified "the results strongly depend on the choice of the turbulence model"

2) Lines 15-21,The sentence beginning with: "The erosion under submarine pipeline can be decomposed into three steps..." is too long. It could be restructured a little bit using "1)", "2)", etc, or ";", or similar.

Response: restructured using a list "1); 2); 3)"

3) Line 83: "(the source code will be available via github after the paper is accepted)" This should be reformulated in the final version, should the manuscript get accepted.

Response: The link to the github repository will replace the sentence between parenthesis

4) Too many "is written", "are written" before equations on page 3. It would be better to vary the style somewhat.

Response: replaced by "given by" and "expressed as"

5) Lines 121 and 126: do not indent the first line after equations, where the quantities used are explained.

Response: indent removed

6) Line 214: "m2.s-2" Put the point in the right position (it appears to have a function of a full-stop here)

Response: point removed

7) Line 322: Correct "One one hand" to "On one hand".

Response: corrected

Reviewer 2 Report

This manuscript is judged as the minor revision and the authors should consider providing more detail about:

What are the limits in your numerical models? More details should be provided in the manuscript. An independent discussion section should be provided in the research paper to discuss your study results.

Author Response

What are the limits in your numerical models? More details should be provided in the manuscript.

Response: the limitation of the two-phase flow model have been detailed in the revised manuscript. The main limitation of the two-phase flow models is the remaining empiricism in the closure models and especially in turbulence models.

An independent discussion section should be provided in the research paper to discuss your study results.

Response: the discussion on the cross diffusion term has been placed in an independent section.